# Mining Synergistic Microbial Interactions: A Roadmap on How to Integrate Multi-Omics Data

**DOI:** 10.3390/microorganisms9040840

**Published:** 2021-04-14

**Authors:** Joao Pedro Saraiva, Anja Worrich, Canan Karakoç, Rene Kallies, Antonis Chatzinotas, Florian Centler, Ulisses Nunes da Rocha

**Affiliations:** 1Department of Environmental Microbiology, Helmholtz Centre for Environmental Research-UFZ, 04318 Leipzig, Germany; joao.saraiva@ufz.de (J.P.S.); anja.worrich@ufz.de (A.W.); canan.karakoc@ufz.de (C.K.); rene.kallies@ufz.de (R.K.); antonis.chatzinotas@ufz.de (A.C.); florian.centler@ufz.de (F.C.); 2German Centre for Integrative Biodiversity Research (iDiv) Halle-Jena-Leipzig, 04103 Leipzig, Germany; 3Institute of Biology, Leipzig University, 04103 Leipzig, Germany

**Keywords:** microbial communities, synergistic interactions, ecosystem processes, multi-omics

## Abstract

Mining interspecies interactions remain a challenge due to the complex nature of microbial communities and the need for computational power to handle big data. Our meta-analysis indicates that genetic potential alone does not resolve all issues involving mining of microbial interactions. Nevertheless, it can be used as the starting point to infer synergistic interspecies interactions and to limit the search space (i.e., number of species and metabolic reactions) to a manageable size. A reduced search space decreases the number of additional experiments necessary to validate the inferred putative interactions. As validation experiments, we examine how multi-omics and state of the art imaging techniques may further improve our understanding of species interactions’ role in ecosystem processes. Finally, we analyze pros and cons from the current methods to infer microbial interactions from genetic potential and propose a new theoretical framework based on: (i) genomic information of key members of a community; (ii) information of ecosystem processes involved with a specific hypothesis or research question; (iii) the ability to identify putative species’ contributions to ecosystem processes of interest; and, (iv) validation of putative microbial interactions through integration of other data sources.

## 1. Introduction

In this review, we discuss a roadmap to mine inter-species interactions in microbial communities. To define this roadmap, we will use an ecosystem process as the unity from which to mine them. Here, we define an ecosystem process as a specific set of metabolic functions (e.g., benzoate degradation or in nitrification). In this review we use the term ecosystem process to define a unit to explore microbial interactions in order to limit microbial community richness to a manageable scale [1]. Currently, a mechanistic understanding of microbial interactions lies on known connections among genes, protein, reactions and their participation in an ecosystem process. Hence, we focus on the synergistic inter-species interactions as these can be directly linked to ecosystem processes. Nevertheless, predicting antagonistic or competitive interactions may be achieved by the inclusion of information such as enzyme kinetics and/or metabolic fluxes [2] involved in these interactions.

## 2. Synergistic Interspecies Interactions Drive Ecosystem Processes

In nature, microbes do not exist alone but rather as members of complex communities [3]. Synergistic interspecies interactions play an essential role in ecosystems by either improving adaptation of microbial communities to their habitats or allowing microorganisms to survive in environments for which they lack the complete metabolic capacity [4]. The type and degree of changes in physicochemical conditions of ecosystems such as the addition of chemicals affect ecosystem processes but also determine how microbial communities respond to those alterations. Species-level functional profiling of (meta)genomes is already possible by providing coverage and abundance estimates on individual pathways across microbial communities as well as for individual species [5]. Expanding this approach to ecosystem processes would generate data allowing the definition of groups of species that cover a full ecosystem process. One first step to determine microbial interactions that are affected by environmental changes (e.g., the introduction of chemicals or temperature shifts) and their relationship to ecosystem functioning relies on the identification and characterization of the constituents of microbial communities as well as their functional potential [6]. 

The concept of “everything is everywhere, but the environment selects” proposed by Baas Becking in 1934 [7] for microbial community assembly has gathered much debate as reviewed by Peter Girguis [8]. Extrapolating this concept to the functional potential (i.e., the complete set of functions) of species, however, is not straightforward. While for core functions such as glucogenesis, the idea that they are genetically widespread in all microorganisms might seem plausible, this is not true for every ecosystem process. For example, methanotrophic bacteria [9] and certain filamentous fungi [6] possess genes that encode a key enzyme in methane degradation—methane monooxygenase—not present in other microorganisms.

Microbial communities will be able to address changing environmental conditions (e.g., introduction of chemicals/substrates, shift in temperature and change in pH) if their combined functional potential encompasses the set of metabolic functions required to handle these changes. Here, functional profile is defined as a subset of metabolic functions from the complete functional pool that are needed for a given ecosystem process (definition adapted from Oh and collaborators [10]). The functional profile can be achieved due to action of single microbes if they are able to solely perform the set of metabolic functions for the required ecosystem process. Alternatively, this functional profile can be achieved by the interaction between two or more species (e.g., acquisition of antibiotic resistance genes via horizontal gene transfer [11], production of secondary metabolites [12] or cross-feeding [13]). It is likely that microbial communities with higher species richness will have a higher number of unique functional traits due to the individual species’ metabolic potential or as the result of the combined metabolic capabilities of multiple species that arises from interspecies interactions. Additionally, microbial community functional potential and functional redundancy is positively correlated with species richness [14]. Fetzer and collaborators [14] also showed that environmental conditions influenced the type and number of microbial interactions (Figure 1). Therefore, more studies characterizing microbial interaction that drive ecosystem processes are necessary as they link microbial diversity and ecosystem function responses to a changing environment [9,14].

### From Where Will One Extract the Genomes to Explore Microbial Interactions?

Currently, advances in single cell sequencing [15], recovery of metagenome assembled genomes [16,17] and advanced cultivation strategies [18] enormously increased the power to identify species and their genomes from natural environments such as soils or deep subsurface. Still, datasets generated from high-throughput sequencing do not provide absolute abundances of species in a microbial community [19] and extra experiments are necessary to generate this data (e.g., quantitative PCR or in situ fluorescence hybridization). Nevertheless, it has been speculated that, in the near future, it will be possible to obtain the genomic information from all species in complex microbial communities [20] thus overcoming one of the bottlenecks in predicting interspecies interactions. Moreover, these approaches have allowed us to gather information on species relative abundance, phylogeny and function [21]. The increase in computational power and advances in sequencing technologies also provides a favorable environment to produce tools and to generate and analyze data in a timewise manner leading to faster and more reliable data mining [22]. These improvements allow the prediction of species interactions and estimation of their functional contribution to ecosystem processes in natural environments, as demonstrated by Kirwan et al. [23]. In the next section, we will discuss advantages and limitations of current approaches to predict interspecies interactions and strategies to overcome them.

## 3. Current Approaches to Predict Microbial Community Functional Profiles and Interspecies Interactions

As stated in the previous section, the functional potential of microbial communities might shed light on potential synergistic interspecies interactions. Several modeling approaches have been proposed to predict microbial community functional profiles [24]. Here, we focus on approaches that take advantage of (meta)genomic data and represent microbial activity in the context of metabolic networks [25] as well as their suitability to infer synergistic interspecies interactions. Three main concepts have been proposed which differ in the level of detail in which they represent the metabolic activity in microbial communities: the supra-organism approach, the population-based approach and the guild approach.

### 3.1. The Supra-Organism Approach

In the supra-organism approach [2], a community is considered as a singular organism and the phylogenetic origin of individual enzymes detected to be present in the community is neglected. Based on metagenomic data, a global metabolic network of enzymatically catalyzed reactions is constructed, allowing for the prediction of shifts in pathway activity when comparing two samples. For example, pathway activity has been inferred in gut metagenomes [26] by calculating gene relative abundance in each individual sample. The supra-organism approach considers interactions between genes (or the respective reactions their enzymes catalyze) rather than between species. This allows the comparison of the functional profiles of microbial communities as whole. However, these models do not allow the prediction of interspecies interactions since the contribution of individual species to any metabolic function is not determined.

### 3.2. The Population-Based Approach

In contrast to the supra-organism approach, species boundaries are explicitly considered in population-based approaches [27]. Here, community members are represented by independent species in the model. These models are especially suited for the analysis of individual species’ functional profile and have been extensively used in metabolic network reconstruction of single genomes [25,28,29]. Furthermore, this approach allows for the inclusion of direct metabolic interactions between community members. The inference of interspecies interactions is possible under this approach since the metabolic contribution of each species to ecosystem processes can be mapped to their genome. However, generating genome-scale metabolic models for all species in a community as well as estimating features such as biomass composition remain challenging [30,31].

### 3.3. The Guild-Based Approach

In the guild-based approach [32], microbial species performing the same metabolic function(s) are grouped together and they are represented by a unique entity in the model, reducing model complexity at the expense of individual species resolution. This approach has been used in ordinary differential equation-based modeling [33] and can be expanded to metabolic network-based approaches [34]. The guild approach is useful when microorganisms are known to possess similar functional traits (e.g. similar methods of organic matter decomposition [35]). However, similar to the supra-organism approach, predicting interspecies interactions in these models is hampered by their inability to identify the contribution of individual species to ecosystem processes [2].

Besides the supra-organism, population-based and guild-based approaches, statistical methods such as Pearson’s and Spearman’s correlation may assist in the identification of interspecies interactions. These methods identify significant relationships by correlating the taxon abundance within microbial communities [36]. However, correlating abundance does not provide information on the underlying mechanisms by which microbes potentially interact. Furthermore, the vast amount of putative interspecies interactions would also require intensive experimental validation.

### 3.4. Advantages and Limitation of Current Approaches Mining Microbial Interactions

As described in Section 3.1, Section 3.2 and Section 3.3, the current approaches used to mine microbial interactions rely on links among genes, enzymes and metabolic reactions. The main limitations and advantages of the different approaches can be found in Table 1. Further, the computational and manual curation efforts in model generation varies strongly as well as the level of detail at which predictions are generated. The supra-organism approach requires the least effort, but only allows for the prediction of general changes in pathways when comparing different samples. Population-based approaches deliver the most detailed quantitative predictions down to intracellular metabolic fluxes of individual species, but require substantial efforts in model generation. These include the generation of genome-scale models and definition of the biomass equation for each community member [37]. Although an ever-increasing number of models become available and semi-automatic pipelines support the generation of novel genome-scale metabolic models, this step often remains difficult to perform. The determination of model parameters regarding cellular maintenance requirements and uptake kinetics are examples of these difficulties. Furthermore, genome scale-based approaches require additional types of data such as transcriptomics, metabolomics and proteomics to validate their results. The inclusion of additional data types would provide information on gene regulation, structure of microbial communities and possibly a link to ecosystem processes. In addition, computational challenges (e.g., random access memory requirements and time) arise in the prediction of putative interspecies interactions together with the increase of species and pathways studied (Equation (1)). For example, for a community of 35 species and a set of 3 reactions, a total of 6545 combinations are possible. A 10-fold increase in the number of species (350) will result in 7,084,700 possible combinations. Expanding the number of reactions to five will yield more than 42 billion possible combinations.
(1)$$C_{k\;}\left(n\right)=\frac{n!}{k!\left(n-k\right)!}$$
where, *n* is the number of species, *k* is the number of reactions and *C* is the number of possible combinations.

Hence, the main limitations to mine microbial interactions can be summarized as:Identification of all species in a community;Incomplete functional annotation of genomes;Data integration and experimental validation; andExponential increase of search space with relatively small increase of number of species or pathway size.

In the next section we discuss the knowns and unknowns in microbial interaction studies through a literature meta-analysis.

## 4. Beyond Genetic Potential: Drawing a Strategy to Mine and Validate Microbial Interactions

Based on the analysis discussed in Section 3.3 and Section 3.4, a road to mine microbial interactions lies in the exploration of the genetic potential involved in specific ecosystem processes. Therefore, we suggest dividing mining of microbial interactions into smaller blocks of information generated from (meta)genomic data. Thus, inference of synergistic interspecies interactions would be determined based on the combined functional potential of all species in a microbial community similar to the study by Jiménez and collaborators [46]. Nevertheless, our meta-analysis indicates that to validate putative microbial interactions extra data is needed (see Section 4.1).

Further, we postulate that prior to identifying microbial interaction it is relevant to focus on specific ecosystem processes. Therefore, estimating the contribution of microbes to ecosystem processes could be determined by looking at its genomic content as long as the connections between genes, proteins and reactions involved in said ecosystem process are known. The identification of a subset of species in a microbial community with the potential, even partial, to participate in an ecosystem process reduces the search space of putative microbial interactions. As discussed in Section 3.4, the reduced search space will make mining relevant inter-species interactions feasible in microbial communities.

In summary, the study of microbial communities is challenging due to our limited capacity to identify all members of a community, to connect genes to proteins and reactions, and to determine species interactions. Hence, mining microbial interactions would include: (**A**) genomic information of all (or key) members of a community; (**B**) information of ecosystem processes involved with a specific hypothesis or research question; and (**C**, **D**) the ability to identify putative species’ contributions to ecosystem processes of interest. A step forward would include standardizing the validation of putative microbial interactions (**G**) through integration of other data sources (**F**). In Section 4.1, we discuss different strategies to validate putative microbial interactions. In Section 4.2, we describe a hypothetical workflow including the mapping of specific ecosystem processes in different members of a microbial community and the validation of putative microbial interactions. In Section 4.3, we provide an example of a study combining omics, data mining and experimental data to infer species contributions to specific ecosystem processes.

### 4.1. Validation of Putative Microbial Interaction through Integration of Different Data Sources

High throughput sequencing has illuminated the black box of microbial diversity. Metagenomics provided an insight into the functional potential of microbial communities without the necessity of cultivating and characterizing thousands of isolates [47,48]. Therefore, metagenomics might provide information on the links between the genetic differences within species and their effects on hosts or adaptability to novel environmental conditions [49]. However, current approaches focusing on predictions of gene functions based on (meta)genomes have four major drawbacks, extensively reviewed by Prosser [50]. First, genetic potential studies may assume that gene presence is directly linked to function [51]. Second, microbial communities are three-dimensional structures that play a crucial role in ecosystem functioning and are not directly assessed by gene presence [52]. Third, different levels of protein activity can be found in different species or among populations of the same species due to transcriptional or post-translational modifications [53]. Fourth, temporal and spatial variability of environmental conditions and community dynamics need to be accounted for when demonstrating microbial interactions [50]. Based on these four limitations, the next four paragraphs discuss strategies that can be added to genetic potential to validate microbial interactions.

#### 4.1.1. Assumption that Gene Presence is Directly Linked to Function

The presence of genetic potential to perform an ecosystem process does not guarantee this process is active [54]. Doolittle and Zhaxybayeva [55] and later Jansson and Hofmockel [56] suggested the investigation of the metaphenome to better understand the functions that are carried out by the active microbial communities under given environmental conditions. The metaphenome considers both microbial functions encoded in the metagenome and biotic and abiotic factors influencing the activity of community members. Thus, it encompasses not only (meta)genomics but also (meta)transcriptomics, (meta)proteomics, metabolomics and factors such as gene silencing. 

#### 4.1.2. Spatial (Three Dimensional) Structure of Microbial Communities

The three-dimensional structure plays an important role in interspecies interactions [57]. Species are not homogeneously distributed within microbial communities but rather structured based on their relationships with each other [58] and are shaped by their metabolic and physiological needs. For example, in biofilms found in suburban bath surfaces such as marble and plaster, growth and microbial community structure are influenced by their susceptibility to light [59]. Heterotrophic microorganisms adapted to dark were mostly found in plaster while low-light adapted microbes were found in mortar [59]. In biological soil crusts bacterial diversity was also shown to be heterogeneous across the different layers [52] and these might influence the number and degree and microbial interactions. Further, in wastewater treatment plant granular sludges presented different growth requirements, community structure and microbial relationships dependent on nitrogen and phosphorous availability [60]. Methods to determine the 3D structure of microbial communities include protein structures [61] and advanced microscopy [57].

#### 4.1.3. Different Levels of Protein Activity within Species or Populations

For an ecosystem process to be active the required genes need to be translated, transcribed and encounter favorable environmental conditions. In cases where the focus is to understand the effect or fate of specific compounds on different ecosystem processes stable isotope probing can be used to trace their transformation and turnover within the different members of a community. Methods such as stable isotope labelling metagenomics [62], metatranscriptomics [63], metaproteomics [64,65] or metabolomics [66] could serve as validation strategies by illustrating metabolite fluxes through groups of predicted interspecies interactions. Other methods such as co-culture experiments could also be employed to validate predicted inter-species interactions [36]. Stable Isotope Probing has also been coupled with NanoSIMS imaging and fluorescent in situ hybridization (FISH) to identify microbial interactions based on the flux of labelled isotopes between microbes [67].

#### 4.1.4. Temporal Variability

It is widely known that microbial communities do not vary only in space but also in time; as deeply studied by the Earth Microbiome Project [68], the NIH Human Project [69] and Tara Oceans [70] among others. A major drawback from most, if not all, techniques to profile microbial communities is that they are based on snapshots of a given community for that specific technique [71]. The distribution of organisms and their functions are not homogeneous and species interactions are not constant in time due to abiotic forces and adaptive mechanisms [72,73,74]. Therefore, to validate patters of microbial interactions one need not only to follow the community over time, but also to have appropriate number of samples in time and space [66,75]. Finally, the time frame for collection of samples and numbers of biological replicates will depend on the biodiversity of a given ecosystem and how dynamic the given process is [76].

When designing strategies to validate microbial interactions, it is necessary to keep in mind that different methods will encompass different limitations and potential levels of detail regarding inter-species microbial interactions (Table 2). For example, studies employing 16S rRNA gene sequencing provide an estimate of the phylogeny of the different species in a microbial community but not their functional potential. Predicting inter-species interactions based on the microbial community’s combined functional potential is in this case not possible. Metabolic network reconstruction of individual genomes from environmental samples can provide substantial detail on the metabolic capabilities of individual species and potential metabolic exchanges. However, the high computational, physiological and chemical information requirements (e.g., biomass equation, reaction reversibility, ATP) to generate reliable genome-scale metabolic networks for such complex environments are seemingly difficult. 

The influence of species on the growth of other community members can also be inferred from co- and mixed-culture experiments by comparing their growth to pure cultures. However, understanding the mechanisms by which species interact requires additional data from complex wet-lab experiments such as stable isotope labelling. A last bottleneck to the validation of interspecies interactions in complex environments is our inability to grow most species in the laboratory [77]. Hence, only a small number of species interactions can be experimentally validated without highly-complex wet-lab procedures.

### 4.2. From Mining to Validation: A Workflow to Identify Mechanisms Underlying Microbial Interactions

In this review, we indicate that defining putative interspecies interactions based solely on genomic potential is only the first piece of the puzzle in gaining a greater in-depth understanding of microbial interactions. Additional information from other data sources and experiments are required (Section 4.1). In this section, we discuss a workflow for iterative mining of microbial interactions by integrating in silico approaches and experimental validation using a thought experiment. In this thought experiment, we use a hypothetical microbial community composed of four species and a hypothetical ecosystem process composed of seven enzymatic reactions (Figure 2). We also assume these reactions require the presence of seven single protein-encoding genes and five protein-encoding genes that participate in the formation of two protein complexes (as you may have multiple proteins involved in the same reaction). 

First, it is necessary to obtain the genomes of all or key members of the microbial community (Figure 2A). Second, a set of genes are selected that represent the reactions involved in the pathway of interest (Figure 2B). Third, genome annotation is performed using only the set of genes defined in the second step (Figure 2C). Fourth, the genetic potential of each species is mapped to the pathway of interest (Figure 2D). This mapping allows to determine which species or groups of species possess the genetic potential for all reactions in the pathway of interest (Figure 2E). Next, additional types of data are integrated in the analysis to validate putative species interactions (Figure 2F). Among others, these can include multi-omics data, such as: (i) (meta)transcriptomics (gene expression), (meta)proteomics (protein expression) and metabolomics (measurement of metabolite production); (ii) three-dimensional structure of the microbial community; (iii) protein abundances measured through stable isotope labelling; (iv) species growth profiles; and, (v) literature and specialized databases. The integration of all these types of data confirms or excludes putative synergistic species interactions (Figure 2G). To note, this process should be iterative as new insights concerning microbial communities and the addition of data to repositories are obtained. For example, genome re-annotation might be necessary if novel species are identified or a better understanding of the reactions that are involved in a specific ecosystem process. 

In the next section we provide an example of a study combining the use of multi-omics, data mining and experimental data. We also provide examples of how their study could be supplemented with additional tools and methods to improve prediction and validation of interspecies interactions.

### 4.3. Assembling a Workflow to Determine Microbial Interactions

Generating and analyzing the data needed to predict microbial interactions focused on specific ecosystem processes can come from a variety of sources such as axenic, enrichment or co-cultures and environmental samples. Additionally, several methods and tools exist to assemble genomes and to assess their quality, to perform functional annotation and to determine the contributions of microbes to ecosystem processes.

**Table 2 microorganisms-09-00840-t002:** Outcomes and limitations of different methods to study microbial interactions. We assigned four validation strategies to confirm microbial interaction as following: (1) Expression or activity assays (e.g., transcriptomics, proteomics, metabolomics, RT-PCR, FBA); (2) 3D structure and spatial variability; (3) Substrate specificity; and, (4) Temporal variability.

Outcome	Limitations	Methods	Environment	Validation				Ref. ^a^
				1	2	3	4	
Improvement in the identification of microbial community species.	Lack of mechanistic understanding of species interactions.	Combination of MALDI-TOF MS ^b^ analysis and high-throughput sequencing 16S rRNA ^c^.	Kimchi	✓	O	O	✓	[78]
		16S rRNA gene sequencing.	Human oral environments	O	✓	O	O	[79]
Demonstration of the influence of abiotic factors on microbial community dynamics.	High computational and data requirements for reconstruction of individual metabolic models.	Metagenomics, metabolic network reconstruction and FBA ^d^.	Anaerobic digestion microbiomes	✓	O	✓	O	[80]
	Lack of mechanistic understanding of species interactions.	PLS-PM ^e^	Rice soil rhizosphere	O	✓	O	✓	[81]
		16S rRNA gene sequencing.	Urban and forest park soil litter layers	O	✓	O	✓	[82]
		In vivo experiment of meadow steppe soil under different precipitation regimes.	Topsoil	✓	✓	O	✓	[83]
	High computational and data requirements for reconstruction of individual metabolic models and complex wet-lab experiments required for validation.	Metabolic network reconstruction, EFM ^f^ and FBA.	Acid-sulfate-chloride springs	✓	O	✓	O	[84]
Demonstration of the influence of interspecies interactions on microbial community dynamics.	Lack of mechanistic understanding of species interactions.	Co-culture of isolates, RNA-Seq ^g^ and RT-qPCR ^h^.	Wine fermentation	✓	O	O	✓	[85]
		qPCR^i^ and 16S rRNA gene sequencing.	Mixed bacterial consortia	✓	O	O	✓	[86]
Improved mechanistic understanding of interspecies interactions.	Complex wet-lab experiments required for validation.	SIP ^j^ and Metagenomics.	Continuous up-flow anaerobic sludge blanket reactors	✓	O	✓	✓	[87]
		Pure and co-cultures and cyclic voltammetry analysis.	Palm oil mill effluent	O	O	✓	✓	[88]
	High computational and data requirements for reconstruction of individual metabolic models.	Mono- and co-culture, metabolic network reconstruction, bipartite graphs, HPLC ^k^, CGQ ^l^, GC-MS ^m^; SIP.	In silicon experiments with pure and co-culture	✓	O	✓	✓	[89]
		Metabolic network reconstruction and cFBA ^n^.	In silicon experiments pure cultures	✓	O	✓	O	[27]
		Metabolic network reconstruction, evolutionary game theory and FBA.	In silicon experiments pure cultures	✓	O	O	O	[90]
		Metagenomics, Metatranscriptomics.	Synthetic human gut	✓	✓	O	O	[5]

^a^ Ref., numbers in between brackets represent references for the different studies; ^b^ MALDI-TOF: matrix-assisted laser desorption/ionization; ^c^ rRNA: Ribosomal ribonucleic acid; ^d^ FBA: Flux Balance Analysis; ^e^ PLS-PM: Partial least squares - path model; ^f^ EFM: elementary flux mode; ^g^ RNA-Seq: Ribonucleic acid sequencing; ^h^ RT-qPCR: Real Time quantitative polymerase chain reaction; ^i^ qPCR: Quantitative polymerase chain reaction; ^j^ SIP: stable isotope probing; ^k^ HPLC: High-performance liquid chromatography; ^l^ CGQ: cell growth quantifier; ^m^ GC-MS: Gas chromatography mass spectrometry; ^n^ cFBA: Community Flux Balance Analysis.

In the next four paragraphs, we will discuss different strategies and tools involved in mining and validating microbial interactions. To foster this discussion, we will analyze the work from Tláskal and collaborators [91]. In this study the authors hypothesize that the decomposition of deadwood required the combined efforts of fungal and bacterial species.

#### 4.3.1. Identifying Microbial Species and Their Genetic Potential

In order to test their hypothesis, Tláskal and collaborators [91] first identified which species were present in deadwood samples. As most soil microorganisms remain uncultured [92], the authors decided to explore the microbial diversity in their system using amplicon metagenomics and metatranscriptomics [91]. Regarding the identification of microbial species, metagenomics was used to recover metagenome-assembled genomes (MAGs) and Metatranscriptomics was used to identify the transcription levels of the small subunit ribosomal RNA. The analysis of small subunit ribosomal RNA transcriptional levels indicate that an organism may be active but does not make a direct link with genetic potential [18]. Although working well for dominant Prokaryotes and DNA viruses, the recovery of MAGs (extensively reviewed by Chen and collaborators [93]) has several limitations for low abundance taxa and Eukaryotes. Other techniques that can be used to identify simultaneously the genetic potential and phylogeny of microbial species encompass single-cell genomics [94] and advanced culturing techniques [95,96].

#### 4.3.2. Defining an Ecosystem Process and Links between Genes, Enzymes and Reactions for a Given Ecosystem Process

Assessing the contribution of species to any given ecosystem process requires functional annotation of the predicted coding sequences to the reactions involved in said process. The National Center for Biotechnology Information (NCBI) [97] is one of the largest repositories for functional annotation [98]. However, mining sequencing data from a large non-specialized database is time-consuming and requires substantial manual curation, as users need to search for all links between genes, enzymes and reactions and to identify and to correct misannotated entries [99]. On the other hand of the spectrum, subject-specific databases are often built upon experimentally validated and/or manually curated data. For example, the Comprehensive Enzyme Information System (BRENDA) [100] collects information on functional enzymes and metabolism. BRENDA’s data includes manual curation obtained from text mining and linked to other curated databases of protein sequences such as the Swiss-Prot [101] from the Uniprot Knowledgebase [102]. In the study by Tláskal and collaborators [91], the authors used one general and two specialized databases for functional annotation. The Kyoto Encyclopedia of Genes and Genomes (KEGG) [103], a non-specialized database, was used to identify all genes in deadwood present in the KEGG database. Using its output, Tláskal and collaborators [91] identified species with the genetic potential to perform methylotrophy and nitrogen cycling. Further, two specialized databases were used to annotate genes involved in carbohydrate utilization in bacteria; respectively, Functional Ontology Assignments for Metagenomes (FOAM) [104] and database of carbohydrate-active enzyme (CAZyme) sequence and annotation (dbCAN HMM) database [105].

#### 4.3.3. Mining Putative Species Interactions

After functional annotation, microbial interactions can be mined by mapping the genetic potential of the different species to each reaction in the selected ecosystem process. In the study by Tláskal and collaborators [91], the contributions of each species to deadwood decomposition was manually assessed by integrating species abundance profiles and relative abundance of genes involved in deadwood decomposition. As a semi-automated alternative, the HMP Unified Metabolic Analysis Network (HumanN2) [5] processes metagenomic and metatranscriptomic data allowing the identification of microbial interactions in complex microbial communities. In the near future, studies involving high-throughput mining of microbial interactions may become broadly used by microbiologists. Therefore, the scientific community is developing automated pipelines such as the Network-Based Tool for Predicting Metabolic Capacities of Microbial Species and their Interactions (NetMet) [106]. NetMet performs automated mining of microbial interactions by linking species-specific enzymatic reactions to metabolites in user-defined environments. In addition, the number of potential combinations can create computational hurdles for mining microbial interactions (Equation (1)). Therefore, limiting the search space is an important step to predict microbial interactions and may be done in silico or in vivo. In silico approaches may consist of correlation analysis or machine learning identification of key microbial species, as previously demonstrated in different environments; such as, coral reefs [107], bioreactors [108] and batch cultures [108]. Current in vivo approaches to deal with the search space involve co-cultures and the use of stable isotopes. Co-cultures limit the search space by controlling the number of species used in the experiments [109]. In addition, stable isotope labeling may limit the search space for the active metabolites [110], proteins [111] or species [112] directly involved in transformations of the labeled compounds. 

#### 4.3.4. Validating Microbial Interactions

As discussed in Section 4.2, validation of microbial interaction may consist of the integration of other data beyond genetic potential. For example, Tláskal and collaborators [91] integrated gene expression patterns obtained from metatranscriptomics and experimentally determined C and N gas fluxes with the metagenomic data to validate the microbial contributions to deadwood decomposition. Microbial interactions were validated when genes mapped to each species and their expression patterns correlated with C and N production rates. However, these links are based on statistical methods (such as correlation analysis) which do not guarantee causality. Furthermore, correlation analysis does not necessarily identify indirect interactions between species [113]. Although integration of metagenomics, metatranscriptomics, CO_2_ production and nitrogen fixation measurements increased substantially the robustness of the predictions, further validation may expand the conclusions achieved by Tláskal and collaborators [91]. For example, studies have shown that spatial structure of microbial communities influence ecosystem processes [114,115,116] and should be taken into account. The 3D structure of the microbial communities could be determined using a molecular ecology network approach as previously demonstrated in biological soil crusts [117]. Tracking the flow of ^13^C-labelled substrates through the metabolism of microbes has also been shown as a valuable method to assess microbial community activity and function [118] even in highly complex and diverse environments [112,119]. Nevertheless, such an approach cannot be carried out in natural forest studies such as the one by Tláskal and collaborators [91].

In summary, each of the different steps from mining to validation of microbial interactions is not limited to a single method or strategy. The selection of approaches to determine microbial interactions will depend on the research questions posed by a study, its source material, the acquisition of the genetic potential for species in a given community and complexity of ecosystem processes of interest.

## 5. Conclusions

Understanding how interspecies interactions contribute to ecosystem functioning is a central issue not only in microbiology but the large field of ecology. Rather than solely depending on diversity measures or correlations, future research should be directed into searching mechanisms underlying causal relationships between community components and their abiotic environments. To optimize the road to uncover interspecies interactions, it is of utmost importance to first identify an ecosystem process of interest. It is relevant to highlight that this ecosystem process should have known connections between genes, proteins and reactions. The next step is to collect genomes of all (or key) species in the microbial community of interest. Mapping of the different reactions to the different genomes is a crucial step as it determines the functional potential of each individual species. This step should be carefully performed since the databases being used to link gene, proteins and reactions are constantly being updated. One must be aware that the size of the pathways involved with the ecosystem process of interest together with the number of species in the respective microbial community has a high-impact on the computational demands to determine microbial interactions and the interpretation of the analysis. In silico correlation analysis and machine learning or in vivo experiments involving co-cultures and stable isotope labeling may circumvent this hurdle. In addition, genetic potential alone is not indicative of microbial interactions. Therefore, the methods needed to validate putative microbial interactions should be chosen based on the complexity of the ecosystem process, the diversity of the microbial community and other biotic and abiotic factors in the ecosystem.

## Figures and Tables

**Figure 1 microorganisms-09-00840-f001:**
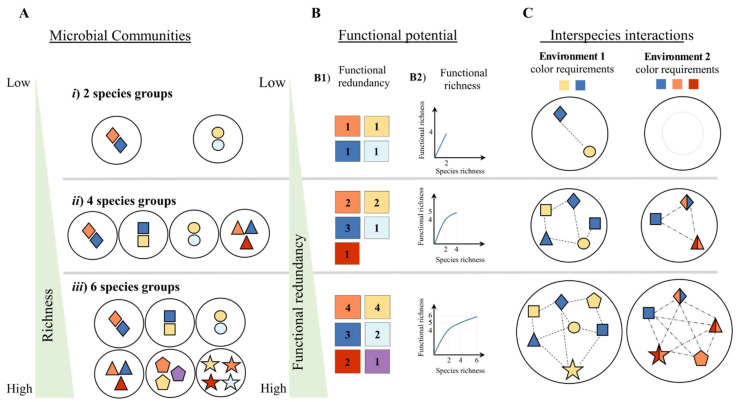
Interplay between microbial community size, functional potential and interspecies interactions. Functional richness and redundancy increase with microbial community richness. Three microbial communities (**A**) are represented with different levels of species richness (*i*, *ii* and *iii*), i.e. different number of unique species (represented by a geometric shape). Each species is capable of performing a number of functions that are represented by a specific color (**B**). For example, community ***i*** is composed by two species each capable of performing two different sets of functions. The number of unique functions illustrates the functional richness of each microbial community. Functional redundancy is determined by the number of microbes with the genetic potential to perform the same function. Thus, an increase in the number of unique species is more likely to result in an increase of the functional redundancy (**B1**) and richness (**B2**) of a microbial community. For example, from community ***ii*** to community ***iii*** there is an increase of two species and one unique function but the number of species capable of performing multiple functions (i.e., functional redundancy) also doubled for four functions: orange, yellow, blue and red. Furthermore, the combinations of interspecies interactions (**C)** is not only dependent on the individual microorganism’s genetic potential but also determined by the environmental conditions. For example, growth in **Environment 1** requires a microbe’s ability to perform two functions (blue and yellow) while for **Environment 2** three functions are required (orange, blue and red). Although not a linear relationship, the higher the number of species in a microbial community, the higher the probability of an increased number of interspecies interactions (as long as the genetic potential is present).

**Figure 2 microorganisms-09-00840-f002:**
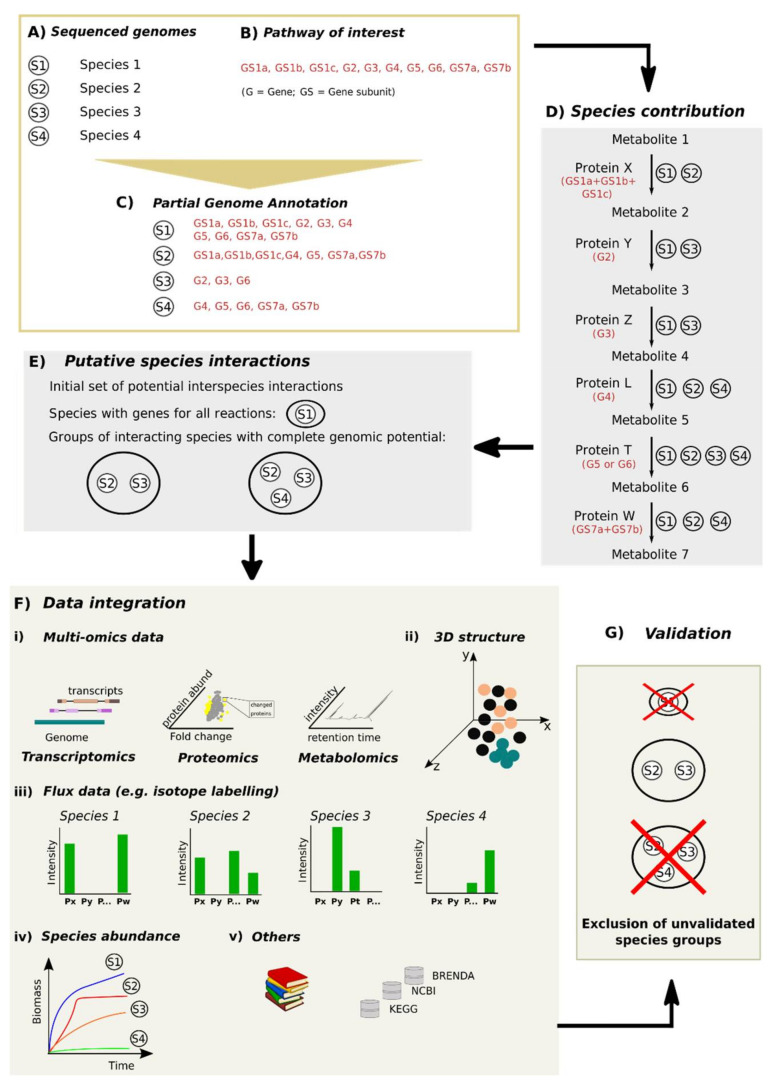
Theoretical work frame for prediction of interspecies interactions. Here, a microbial community is composed of four species (**A**) and a hypothetical ecosystem process/pathway of interest (**B**) requires the presence of five single protein-encoding genes and five protein-encoding genes that participate in the formation of two protein complexes. First, annotation of individual genomes from the microbial community is performed using only the set of target genes (**C**). From here, one can determine which species possess the complete functional potential to perform the target pathway (**D**). Additionally, one can also determine which species or groups of species possess a combined genomic potential to perform the complete ecosystem process (**E**) putative interacting species. Further refinement of the generated lists can be achieved by the inclusion of experimental data, species absolute abundances, literature searches, specialized databases and other omics data types (e.g., transcriptomics, metabolomics and proteomics) (**F**) leading to increased robustness of predictions and reduction of the number of interspecies interactions for experimental validation (**G**).

**Table 1 microorganisms-09-00840-t001:** Pros and cons of current modelling approaches to predict microbial interactions, environments where the selected approaches have been used and respective references.

Approach	Pros	Cons	Environments	References
Supra-organism	Global reaction network is possible and allows for prediction of shifts in pathway activity by measuring gene relative abundance.	Genetic potential of individual species not determined.	Anaerobic mixed culture fermentations	[38]
		Contribution of individual species to shifts in pathway activity not determined since interactions are based on genes/reactions.	Agricultural soil and seep sea “whale fall” carcasses	[39]
Population-based	Species boundaries explicitly defined. Individual species functional potential can be determined. Allows determining direct metabolic interactions between species.	High computational and manual curation efforts since full genome-scale metabolic models for each species is required.	Corals	[40]
			Anoxic sediments	[41]
			Batch and Continuous cultures	[42]
			Synthetic microbial systems	[43]
Guild-based	Less complex models since grouping of species is based on their known functional traits.	Requires previous knowledge on functional traits. Individual contribution of species to ecosystem processes is unknown.	Soil	[44,45]

## Data Availability

Not applicable.

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
