# Peer review of "Mining Synergistic Microbial Interactions: A Roadmap on How to Integrate Multi-Omics Data"

_microorganisms, 2021, doi:10.3390/microorganisms9040840_

Round 1
Reviewer 1 Report
In their study "Mining synergistic microbial interactions: a roadmap on how to integrate multi-omics data" the authors are reviewing the current state of data evaluation in metagenomics and make recommendations how to handle data to understand interactions within a microbial community in a better way. Though the study is well written and provides some overview of the current state (according to the references), the study is in most of the parts quite general, and does not (yet) provide an applicable "roadmap", rather than more superficial clues.
There are many ways to give a more concrete description: e.g. application examples to be discussed in more detail, description of data mining processes (including applicable software) and more detailed evaluations of examples as given in table 1. In its current state, it is left to the reader, based on the references given, to solve the problem of efficient data mining of microbial communities.
specific comments:
line 14: what are "building blocks" here?
line 32: "we use ecosystem process..." this ecosystem process? ecosystem processes?
line 108/109: "... are limited in their ability to provide ..." provide limited information on...
line 231 ff: "Hence, a step forward in mining microbial interactions would include: (A) ..." I would agree with the items listed, but to me, this does not appear "a step forward" but is rather the way how currently the data mining is done.
line 266-277: in this section I would rather expect examples from soil and/or sediment microbial comunities, not just because of their complex 3D patterns, but due to their overall relevance in ecosystems worldwide.
formatting of tables is a bit odd: flush left
Author Response
Point by point replies to comments by the Referee 1
- Comment 1
Though the study is well written and provides some overview of the current state (according to the references), the study is in most of the parts quite general, and does not (yet) provide an applicable "roadmap", rather than more superficial clues. There are many ways to give a more concrete description: e.g. application examples to be discussed in more detail, description of data mining processes (including applicable software) and more detailed evaluations of examples as given in table 1. In its current state, it is left to the reader, based on the references given, to solve the problem of efficient data mining of microbial communities.
Reply: We would like to thank Referee 1 for the feedback and suggestions and have now added a new section (4.3. Assembling a workflow to determine microbial interactions). In this new section we provide a study case deconstructed into 4 paragraphs describing the: 1) different methods employed to identify species, 2) databases used for functional annotation, 3) methods for mining of microbial interactions and 4) validation strategies employed. In each paragraph, we also provide the reader with an explanation on the advantages and limitations of the used tools and suggest alternatives. We believe that the inclusion of this new section provides the reader with a more concrete description of the different steps for efficient data mining of microbial interactions.
- Comment 2
Line 14: what are "building blocks" here?
Reply 2: We would like to thank Referee 1 for the observation. In line 14 we wanted to highlight the use of genetic potential as the starting point for mining microbial interactions. To improve clarity of the statement we have changed the text to “as a starting point”. The modified text can be found as written below in the updated version of the manuscript:
Line 14: “Nevertheless, it can be used as the starting point to infer synergistic (…)”
- Comment 3
line 32: "we use ecosystem process..." this ecosystem process? ecosystem processes?
Reply 3: We would like to thank Referee 1 for the observation. Indeed the use of ecosystem process and variations could confuse the reader. We have made changes to lines 31 and 32 to prevent confusion. The modified text can be found as written below in the updated version of the manuscript:
Lines 31-32: “Here, we define an ecosystem process as a specific set of metabolic functions (e.g. benzoate degradation or in nitrification). In this review we use the term ecosystem process to define a unit to explore microbial interactions in order (…)”.
- Comment 4
line 108/109: "... are limited in their ability to provide ..." provide limited information on...
Reply 4: We would like to thank Referee 1 for the suggestion. The modified text can be found as written below in the updated version of the manuscript:
Lines 110-113: “Still, datasets generated from high-throughput sequencing do not provide absolute abun-dances of species in a microbial community [19] and extra experiments are necessary to generate this data (e.g., quantitative PCR or in situ fluorescence hybridization).”.
- Comment 5
line 231 ff: "Hence, a step forward in mining microbial interactions would include: (A) ..." I would agree with the items listed, but to me, this does not appear "a step forward" but is rather the way how currently the data mining is done.
Reply 5: We would like to thank Referee 1 for the observation. We have modified the text to clearly state that the “step forward” in mining microbial interactions is the standardization of validation strategies by integrating additional data types. We also bold the different letters corresponding to the different steps to make them easier for the reader to follow. The modified text can be found as written below in the updated version of the manuscript:
Line 235: “A step forward would include standardizing the validation of putative microbial interactions (G) through integration of other data sources (F).”
- Comment 6
line 266-277: in this section I would rather expect examples from soil and/or sediment microbial communities, not just because of their complex 3D patterns, but due to their overall relevance in ecosystems worldwide.
Reply 6: We would like to thank Referee 1 for the suggestion. We have added an additional example of biological soil crusts in section 4.1. The modified text can be found as written below in the updated version of the manuscript:
Line 280-282: “In biological soil crusts bacterial diversity was also shown to be heterogeneous across the different layers [52] and these might influence the number and degree and microbial interactions.”
- Comment 7
formatting of tables is a bit odd: flush left
Reply 7: We would like to thank Referee 1 for the observation. We have formatted the tables 1 and 2 to improve readability. Formatting included the addition of spaces between lines and columns, indenting of text to distinguish between paragraphs, removing bullet points and aligning text to the top left.
Reviewer 2 Report
The review discusses the different approaches used for understanding the microbial interactions, their limitations and outcomes and therefore is highly important. Table 2 summarize four validation strategies of various methods and allows the excellent orientation in drawbacks of different approaches. I suggest, that in Conlusion should be also emphasized the methods like SIP and co-cultures experiments because these aproaches allow us to understand the interspecies interactions without high computational demands. Other comments: Unify the Table 1 and Table 2. Do not use bullets in Table 2 and remove the full stop. Table 2 - correct “High computational and data requirements for reconstruction of individual metabolic models and complex wet-lab experiments required for validation“Author Response
Reply to comments by the Referee 2
- Comment 1
The review discusses the different approaches used for understanding the microbial interactions, their limitations and outcomes and therefore is highly important. Table 2 summarize four validation strategies of various methods and allows the excellent orientation in drawbacks of different approaches. I suggest, that in Conclusion should be also emphasized the methods like SIP and co-cultures experiments because these aproaches allow us to understand the interspecies interactions without high computational demands.
.
Reply 1: We would like to thank Referee 2 for the suggestions. We agree with the reviewer that the use of SIP and co-cultures are computational-independent methods with the potential to infer microbial interactions. However, mining and validating microbial interactions in highly complex natural communities is unfeasible without the reduction of the search space. The use of SIP and co-culture experiments is thus advised after this procedure as mentioned in a new section added to the manuscript (4.3 Assembling a workflow to determine microbial interactions – Validating microbial interactions). Additionally, we have added text in the Conclusion of the updated version of the manuscript indicating the possibility of using co-cultures and stable isotope labelling methods to circumvent the high search space of potential interspecies interactions in complex communities. The modified text in the Conclusion can be found as written below in the updated version of the manuscript:
Lines 493-495: “In silico correlation analysis and machine learning or in vivo experiments involving co-cultures and stable isotope labeling may circumvent this hurdle.”
- Comment 2
Other comments: Unify the Table 1 and Table 2.
Reply 2: We would like to thank Referee 2 for the suggestion. Our aim in Table 1 was to present the reader with the current main approaches in the study of microbial communities and their respective advantages and limitations. Table 2 was constructed to provide the reader with actual studies, methods, limitations in mining microbial interactions and possible strategies for validation of microbial interactions based on the type of data generated. Therefore, we believe the two tables are necessary to make the text more clear to the reader.
- Comment 3
Do not use bullets in Table 2 and remove the full stop.
Reply 3: We would like to thank Referee 2 for the suggestion. We have formatted the tables 1 and 2 to improve readability. Formatting included the addition of spaces between lines and columns, indenting of text to distinguish between paragraphs, removing bullet points and aligning text to the top left.
- Comment 4
Table 2 - correct “High computational and data requirements for reconstruction of individual metabolic models and complex wet-lab experiments required for validation“
Reply 4: We would like to thank Referee 2 for the observation. We have modified the text in Table 2 as written below in the updated version of the manuscript:
“High computational and data requirements for reconstruction of individual metabolic models and complex wet-lab experiments required for validation.”